# Meta Dropout: Learning to Perturb Latent Features for Generalization

**Hae Beom Lee**[1]**, Taewook Nam**[1]**, Eunho Yang**[1,2]**, Sung Ju Hwang**[1,2]
KAIST[1], AITRICS[2], South Korea
{haebeom.lee,namsan,eunhoy,sjhwang82}@kaist.ac.kr

## Abstract

A machine learning model that generalizes well should obtain low errors on unseen test examples. Thus, if we know how to optimally perturb training examples to account for test examples, we may achieve better generalization performance. However, obtaining such perturbation is not possible in standard machine learning frameworks as the distribution of the test data is unknown. To tackle this challenge, we propose a novel regularization method, *meta-dropout*, which *learns to perturb* the latent features of training examples for generalization in a meta-learning framework. Specifically, we meta-learn a noise generator which outputs a multiplicative noise distribution for latent features, to obtain low errors on the test instances in an input-dependent manner. Then, the learned noise generator can perturb the training examples of unseen tasks at the meta-test time for improved generalization. We validate our method on few-shot classification datasets, whose results show that it significantly improves the generalization performance of the base model, and largely outperforms existing regularization methods such as information bottleneck, manifold mixup, and information dropout.

## 1 Introduction

Obtaining a model that generalizes well is a fundamental problem in machine learning, and is becoming even more important in the deep learning era where the models may have tens of thousands of parameters. Basically, a model that generalizes well should obtain low error on unseen test examples, but this is difficult since the distribution of test data is unknown during training. Thus, many approaches resort to variance reduction methods, that reduce the model variance with respect to the change in the input. These approaches include controlling the model complexity (Neyshabur et al., 2017), reducing information from inputs (Tishby et al., 1999), obtaining smoother loss surface (Shirish Keskar et al., 2017; Neyshabur et al., 2017; Chaudhari et al., 2017; Santurkar et al., 2018), smoothing softmax probabilities (Pereyra et al., 2017) or training for multiple tasks with multi-task (Caruana, 1997) and meta-learning (Thrun & Pratt, 1998).

A more straightforward and direct way to achieve generalization is to *simulate* the test examples by perturbing the training examples during training. Some regularization methods such as mixup (Zhang et al., 2017) follow this approach, where the training examples are perturbed to the direction of the other training examples to mimic test examples. The same method could be also applied to the latent feature space, to achieve even larger performance gain (Verma et al., 2019). However, these approaches are all limited in that they do not explicitly aim to lower the generalization error on the test examples. How can we then perturb the training instances such that the perturbed instances will be actually helpful in lowering the test loss? Enforcing this generalization objective is not straightforward in standard learning framework since the test data is unobservable.

To solve this seemingly impossible problem, we resort to meta-learning (Thrun & Pratt, 1998) which aims to learn a model that generalize over a distribution of tasks, rather than a distribution of data instances from a single task. Generally, a meta-learner is trained on a series of tasks with random training and test splits. While learning to solve diverse tasks, it accumulates the meta-knowledge that is not specific to a single task, but is generic across all tasks, which is later leveraged when learning for a novel task. During this meta-training step, we observe both the training and test data. That is, we can explicitly learn to perturb the training instances to obtain low test loss in this meta-learning

framework. The learned noise generator then can be used to perturb instances for generalization at meta-test time.

Yet, learning how much and which features to perturb is difficult for two reasons. First of all, meaningful directions of perturbation may differ from one instance to another, and one task to another. Secondly, a single training instance may need to cover largely different test instances with its perturbation, since we do not know which test instances will be given at test time. To handle this problem, we propose to learn an input-dependent stochastic noise; that is, we want to learn distribution of noise, or perturbation, that is meaningful for a given training instance. Specifically, we learn a noise generator for each layer features of the main network, given lower layer features as input. We refer to this meta-noise generator as *meta-dropout* that *learns to regularize*.

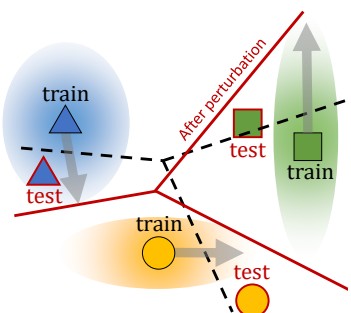

Figure 1: **Concepts.** In the feature space, each training instance stochastically perturbs so that the resultant decision boundaries (red line) explain well for the test examples. Note that the noise distribution does not have to cover the test instances directly.

Also the learned noise distribution is transferable, which is especially useful for few-shot learning setting where only a few examples are given to solve a novel task. Figure 1 depicts such a scenario where the noise generator perturbs each input instance to help the model predict better decision boundaries.

In the remaining sections, we will explain our model in the context of existing work and propose the learning framework for meta-dropout. We compare our method to existing regularizers such as manifold mixup (Verma et al., 2019), information bottleneck, and information dropout (Achille & Soatto, 2018), which our method significantly outperforms. We further show that meta-dropout can be understood as meta-learning the variational inference framework for the graphical model in Figure 3. Finally, we validate our work on multiple benchmark datasets for few-shot classification.

Our contribution is threefold.

- We propose a novel regularization method called *meta-dropout* that generates stochastic input-dependent perturbations to regularize few-shot learning models, and propose a meta-learning framework to train it.
- We compare with the existing regularizers such as information bottleneck (Achille & Soatto, 2018; Alemi et al., 2017) and manifold mixup (Verma et al., 2019). We also provide the probabilistic interpretation of our approach as learning to regularize the variational inference framework for the graphical model in Figure 3.
- We validate our method on multiple benchmark datasets for few-shot classification, on which our model significantly improves the performance of the base models.

## 2 RELATED WORK

**Meta learning** While the literature on meta-learning (Thrun & Pratt, 1998) is vast, here we discuss a few relevant existing works for few-shot classification. One of the most popular approaches is metric-based meta-learning that learns a shared metric space (Koch et al., 2015; Vinyals et al., 2016; Snell et al., 2017; Oreshkin et al., 2018; Mishra et al., 2018) over randomly sampled few-shot classification problems, to embed the instances to be closer to their correct embeddings by some distance measure regardless of their classes. The most popular models among them are Matching networks (Vinyals et al., 2016) which leverages cosine distance measure, and Prototypical networks (Snell et al., 2017) that make use of Euclidean distance. On the other hand, gradient-based approaches (Finn et al., 2017; 2018; Li et al., 2017; Lee & Choi, 2018; Ravi & Beatson, 2019; Zintgraf et al., 2019) learns a shared initialization parameter, which can rapidly adapt to new tasks with only a few gradient steps. Recent literatures on few-shot classification show that the performance can significantly improve with larger networks, meta-level regularizers or fine-tuning (Lee et al., 2019; Rusu et al., 2019). Lastly, meta-learning of the regularizers has been also addressed in Balaji et al. (2018), which proposed to meta-learn $\ell_1$ regularizer for domain adaptation. However, while this work focuses on the meta-learning of the hyperparameter of generic regularizers, our model is more explicitly targeting generalization via input perturbation.

**Dropout**   Dropout (Srivastava et al., 2014) is a regularization technique to randomly drop out neurons during training. In addition to feature decorrelation and ensemble effect, we could also interpret dropout regularization as a variational approximation for posterior inference of the network weights (Gal & Ghahramani, 2016), in which case we can even learn the dropout rates with stochastic gradient variational Bayes (Kingma et al., 2015; Gal et al., 2017). The dropout regularization could be viewed as a noise injection process. In case of standard dropout, the noise follows the Bernoulli distribution, but we could also use Gaussian multiplicative noise to the parameters instead (Wang & Manning, 2013). It is also possible to learn the dropout probability in an input-dependent manner as done with Adaptive Dropout (Standout) (Ba & Frey, 2013), which could be interpreted as variational inference on input-dependent latent variables (Sohn et al., 2015; Xu et al., 2015; Heo et al., 2018; Lee et al., 2018). However, meta-dropout is fundamentally different from adaptive dropout, as it makes use of previously obtained meta-knowledge in posterior variance inference, while adaptive dropout resorts only to training data and prior distribution.

**Regularization methods**   There exist large number of regularization techniques for improving the generalization performance of deep neural networks (Srivastava et al., 2014; Ioffe & Szegedy, 2015; Ghiasi et al., 2018), but we only discuss approaches based on input-dependent perturbations that are closely related to meta-dropout. Mixup (Zhang et al., 2017) randomly pairs training instances and interpolates between them to generate additional training examples. Verma et al. (2019) further extends the technique to perform the same procedure in the latent feature spaces at each layer of deep neural networks. While mixup variants are related to meta-dropout as they generate additional training instances via input perturbations, these heuristics may or may not improve generalization, while meta dropout perturbs the input while explicitly aiming to minimize the test loss in a meta-learning framework. Information-theoretic regularizers (Alemi et al., 2017; Achille & Soatto, 2018) are also relevant to our work, where they inject input-dependent noise to latent features to forget some of the information from the inputs, resulting in learning high-level representations that are invariant to less meaningful variations. Yet, meta-dropout has a more clear and direct objective to improve on the generalization. Adversarial learning (Goodfellow et al., 2015) is also somewhat related to our work, which perturbs the examples towards the direction that maximizes the training loss, as meta dropout also tend to pertub inputs to the direction of decision boundaries. In the experiments section, we show that meta dropout also improves adversarial robustness, which implies the connection between adversarial robustness and generalization (Stutz et al., 2019).

## 3   LEARNING TO PERTURB LATENT FEATURES

We now describe our problem setting and the meta-learning framework for learning to perturb training instances in the latent feature space, for improved generalization. The goal of meta-learning is to learn a model that generalizes over a task distribution $p(\mathcal{T})$. This is usually done by training the model over large number of tasks (or episodes) sampled from $p(\mathcal{T})$, each of which consists of a training set $\mathcal{D}^{\text{tr}} = \{(\mathbf{x}_i^{\text{tr}}, \mathbf{y}_i^{\text{tr}})\}_{i=1}^N$ and a test set $\mathcal{D}^{\text{te}} = \{(\mathbf{x}_j^{\text{te}}, \mathbf{y}_j^{\text{te}})\}_{j=1}^M$.

Suppose that we are given such a split of $\mathcal{D}^{\text{tr}}$ and $\mathcal{D}^{\text{te}}$. Denoting the initial model parameter of an arbitrary neural network as $\boldsymbol{\theta}$, Model Agnostic Meta Learning (MAML) (Finn et al., 2017) aims to infer task-specific model parameter $\boldsymbol{\theta}^*$ with one or a few gradient steps with the training set $\mathcal{D}^{\text{tr}}$, such that $\boldsymbol{\theta}^*$ can quickly generalize to $\mathcal{D}^{\text{te}}$ with a few gradient steps. Let $\alpha$ denote the inner-gradient step size, $\mathbf{X}$ and $\mathbf{Y}$ denote the concatenation of input data instances and their associated labels respectively for both training and test set. Then, we have

$$\min_{\boldsymbol{\theta}} \mathbb{E}_{p(\mathcal{T})} \left[ -\frac{1}{M} \log p(\mathbf{Y}^{\text{te}} | \mathbf{X}^{\text{te}}; \boldsymbol{\theta}^*) \right], \quad \text{where} \quad \boldsymbol{\theta}^* = \boldsymbol{\theta} - \alpha \nabla_{\boldsymbol{\theta}} \left( -\frac{1}{N} \log p(\mathbf{Y}^{\text{tr}} | \mathbf{X}^{\text{tr}}; \boldsymbol{\theta}) \right). \quad (1)$$

Optimizing the objective in Eq. 1 is repeated over many random splits of $\mathcal{D}^{\text{tr}}$ and $\mathcal{D}^{\text{te}}$, such that the initial model parameter $\boldsymbol{\theta}$ captures the most generic information over the task distribution $p(\mathcal{T})$.

### 3.1   META-DROPOUT

A notable limitation of MAML is that the knowledge transfer to unseen tasks is done only by sharing the initial model parameter over the entire task distribution. When considering few-shot classification task for instance, this means that given the initial $\boldsymbol{\theta}$, the inner-gradient steps at the meta-test

time only depends on few training instances, which could potentially lead to learning sub-optimal decision boundaries. Thus, it would be desirable if we could transfer additional information from the meta-learning step, that could help the model to generalize better. Based on this motivation, we propose to capture a transferable noise distribution from the given tasks at the meta-training time, and inject the learned noise at the meta-test time, such that the noise samples would perturb the latent features of the training instances to explicitly improve the decision boundaries.

Toward this goal, we propose to learn an input-dependent noise distribution, such that the noise is individually tailored for each instance. This is because the optimal direction and the amount of perturbation for each input instance may vary largely from one instance to another. We empirically validate the effectiveness of this input-dependent noise generation in our experiments (Table 3). We denote the shared form of the noise distribution as $p(\mathbf{z}|\mathbf{x}^{\text{tr}}; \boldsymbol{\theta}, \boldsymbol{\phi})$, with $\boldsymbol{\phi}$ as the additional meta-parameter for the noise generator that does not participate in the inner-optimization. Note that the noise distribution has dependency on the main parameter $\boldsymbol{\theta}$, due to input dependency (see Figure 2).

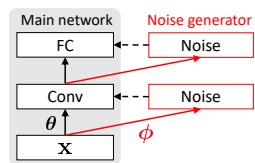

Figure 2: **Model architecture.** Each bottom layer generates the noise for upper layer with parameter $\boldsymbol{\phi}$.

Recall that each inner-gradient step requires to compute the gradient over the training marginal log-likelihood (Eq. 1). In our case of training with the input-dependent noise generator $p(\mathbf{z}|\mathbf{x}^{\text{tr}}; \boldsymbol{\theta}, \boldsymbol{\phi})$, the marginal log-likelihood is obtained by considering all the plausible perturbations for each instance: $\log p(\mathbf{Y}^{\text{tr}}|\mathbf{X}^{\text{tr}}; \boldsymbol{\theta}, \boldsymbol{\phi}) = \sum_{i=1}^{N} \log \mathbb{E}_{\mathbf{z}_i \sim p(\mathbf{z}_i|\mathbf{x}_i^{\text{tr}}; \boldsymbol{\theta}, \boldsymbol{\phi})} [p(\mathbf{y}_i^{\text{tr}}|\mathbf{x}_i^{\text{tr}}, \mathbf{z}_i; \boldsymbol{\theta})]$. In this work, we instead consider its lower bound, which simply corresponds to the expected loss over the noise distribution (See section 3.2 for more discussion).

$$\log p(\mathbf{Y}^{\text{tr}}|\mathbf{X}^{\text{tr}}; \boldsymbol{\theta}, \boldsymbol{\phi}) \geq \sum_{i=1}^{N} \mathbb{E}_{\mathbf{z}_i \sim p(\mathbf{z}_i|\mathbf{x}_i^{\text{tr}}; \boldsymbol{\theta}, \boldsymbol{\phi})} \left[ \log p(\mathbf{y}_i^{\text{tr}}|\mathbf{x}_i^{\text{tr}}, \mathbf{z}_i; \boldsymbol{\theta}) \right] \tag{2}$$

We take the gradient ascent steps with this lower bound. Following Kingma & Welling (2014), we use reparameterization trick to evaluate the gradient of the expectation w.r.t. $\boldsymbol{\theta}$ and $\boldsymbol{\phi}$, such that $\mathbf{z} = \text{Softplus}(\boldsymbol{\mu} + \boldsymbol{\varepsilon})$ and $\boldsymbol{\varepsilon} \sim \mathcal{N}(\mathbf{0}, \mathbf{I})$ (See Eq. 5). Then, the associated Monte-carlo (MC) samples allow to compute the gradients through the deterministic function $\boldsymbol{\mu}$ parameterized by $\boldsymbol{\phi}$ and $\boldsymbol{\theta}$. We use MC sample size $S = 1$ for meta-training and $S = 30$ for meta-testing.

$$\boldsymbol{\theta}^* = \boldsymbol{\theta} + \alpha \frac{1}{N} \sum_{i=1}^{N} \frac{1}{S} \sum_{s=1}^{S} \nabla_{\boldsymbol{\theta}} \log p(\mathbf{y}_i^{\text{tr}}|\mathbf{x}_i^{\text{tr}}, \mathbf{z}_i^{(s)}; \boldsymbol{\theta}), \quad \mathbf{z}_i^{(s)} \overset{\text{i.i.d.}}{\sim} p(\mathbf{z}_i|\mathbf{x}_i^{\text{tr}}; \boldsymbol{\phi}, \boldsymbol{\theta}) \tag{3}$$

By taking the proposed gradient step, the target learning process can consider all the plausible perturbations of the training examples that can help explain the test dataset. Extension to more than one inner-gradient step is also straightforward: for each inner-gradient step, we perform MC integration to estimate the model parameter at the next step, and repeat this process until we get the final $\boldsymbol{\theta}^*$.

Finally, we evaluate and maximize the performance of $\boldsymbol{\theta}^*$ on the test examples, by optimizing $\boldsymbol{\theta}$ and $\boldsymbol{\phi}$ for the following meta-objective over the task distribution $p(\mathcal{T})$. Considering that the test examples should remain as stable targets we aim to generalize to, we deterministically evaluate its log-likelihood by forcing the variance of $\mathbf{a}$ to be zero in Eq. 5 (i.e. $\mathbf{a} = \boldsymbol{\mu}$). Denoting $\bar{\mathbf{z}}$ as the one obtained from such deterministic $\mathbf{a}$, we have

$$\max_{\boldsymbol{\theta}, \boldsymbol{\phi}} \mathbb{E}_{p(\mathcal{T})} \left[ \frac{1}{M} \sum_{i=1}^{M} \log p(\mathbf{y}_i^{\text{te}}|\mathbf{x}_i^{\text{te}}, \mathbf{z}_i = \bar{\mathbf{z}}_i; \boldsymbol{\theta}^*) \right]. \tag{4}$$

By applying the same deterministic transformation $\boldsymbol{\mu}$ to both training and test examples, they can share the same consistent representation space, which seems important for performance. Lastly, to compute the gradient of Eq. 4 w.r.t. $\boldsymbol{\phi}$, we must compute second-order derivative, otherwise the gradient w.r.t. $\boldsymbol{\phi}$ will always be zero. See Algorithm 1 and 2 for the pseudocode of Meta-dropout.

**Form of the noise** We apply input-dependent multiplicative noise to the latent features at all layers (Ba & Frey, 2013) (See Figure 2). Here we suppress the dependency on $\boldsymbol{\theta}$ and $\boldsymbol{\phi}$ for better readability. We propose to use simple Softplus transformation of a Gaussian noise distribution. We

could use other types of transformations, such as exponential transformation (i.e. Log-Normal distribution), but we empirically verified that Softplus works better. First, we generate input-dependent noise $\mathbf{z}^{(l)}$ given the latent features $\mathbf{h}^{(l-1)}$ from the previous layer.

$$\mathbf{z}^{(l)} = \text{Softplus}(\mathbf{a}^{(l)}), \quad \mathbf{a}^{(l)} \sim \mathcal{N}(\mathbf{a}^{(l)}|\boldsymbol{\mu}^{(l)}, \mathbf{I}) \tag{5}$$

where $\boldsymbol{\mu}^{(l)} := \boldsymbol{\mu}^{(l)}(\mathbf{h}^{(l-1)})$ is parameterized by $\phi$ and $\boldsymbol{\theta}$. Note that although the variance of $\mathbf{a}$ is fixed as $\mathbf{I}$, we can still adjust the noise scale at each dimension via the mean $\boldsymbol{\mu}^{(l)}$, with the scale of the noise suppressed at certain dimensions by the Softplus function. While we could also model the variance in an input-dependent manner, we empirically found that this does not improve the generalization performance (Table 3). Then, we can obtain the latent features $\mathbf{h}^{(l)}$ for the current layer $l$, by applying the noise to the pre-activation $\mathbf{f}^{(l)} := \mathbf{f}^{(l)}(\mathbf{h}^{(l-1)})$ parameterized by $\boldsymbol{\theta}$:

$$\mathbf{h}^{(l)} = \text{ReLU}(\mathbf{f}^{(l)} \circ \mathbf{z}^{(l)}). \tag{6}$$

where $\circ$ denotes the element-wise multiplication.

## 3.2 Connection to Variational Inference

Lastly, we explain the connection between the lower bound in Eq. 2 and the variational inference with the graphical model in Figure 3. Suppose we are given a training set $\mathcal{D} = \{(\mathbf{x}_i, \mathbf{y}_i)\}_{i=1}^{N}$, where for each instance $\mathbf{x}_i$, the generative process involves latent $\mathbf{z}_i$ conditioned on $\mathbf{x}_i$. Note that during the inner-gradient steps, the global parameter $\phi^*$ is fixed and we only optimize $\boldsymbol{\theta}$. Based on this specific context, we derive the lower bound $L(\boldsymbol{\theta})$ as follows:

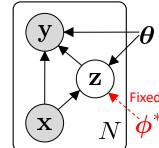

Figure 3: Graphical model

$$\log p(\mathbf{Y}|\mathbf{X}; \boldsymbol{\theta}, \phi^*)$$
$$\geq \sum_{i=1}^{N} \mathbb{E}_{q(\mathbf{z}_i|\mathbf{x}_i, \mathbf{y}_i)}[\log p(\mathbf{y}_i|\mathbf{x}_i, \mathbf{z}_i; \boldsymbol{\theta})] - \text{KL}[q(\mathbf{z}_i|\mathbf{x}_i, \mathbf{y}_i)\|p(\mathbf{z}_i|\mathbf{x}_i; \boldsymbol{\theta}, \phi^*)] \tag{7}$$
$$= \sum_{i=1}^{N} \mathbb{E}_{p(\mathbf{z}_i|\mathbf{x}_i; \boldsymbol{\theta}, \phi^*)}[\log p(\mathbf{y}_i|\mathbf{z}_i, \mathbf{x}_i; \boldsymbol{\theta})] \tag{8}$$
$$= L(\boldsymbol{\theta}).$$

where from Eq. 7 to Eq. 8 we let the approximate posterior $q(\mathbf{z}|\mathbf{x}, \mathbf{y})$ share the same form with the conditional prior $p(\mathbf{z}|\mathbf{x}; \boldsymbol{\theta}, \phi^*)$, such that the KL divergence between them becomes zero. By sharing the same form, we can let the training and testing pipeline be consistent (note that the label $\mathbf{y}$ is not available for the test examples, see Sohn et al. (2015) and Xu et al. (2015)). By maximizing $L(\boldsymbol{\theta})$, we obtain the task-specific model parameter $\boldsymbol{\theta}^*$ that can make more accurate predictions on the test examples, owing to the knowledge transfer from $\phi^*$, which regularizes the final form of the conditional prior $p(\mathbf{z}|\mathbf{x}; \boldsymbol{\theta}^*, \phi^*)$.

## 4 Experiments

We now validate our meta-dropout on few-shot classification tasks for its effectiveness.

**Baselines and our models** We first introduce the two most important baselines and our model. We compare against other few-shot meta-learning baselines, using their reported performances.
**1) MAML.** Model Agnostic Meta Learning by Finn et al. (2017). First-order approximation is not considered for fair comparison against the baselines that use second-order derivatives.
**2) Meta-SGD.** A variant of MAML whose learning rate vector is element-wisely learned for the inner-gradient steps (Li et al., 2017).
**3) Meta-dropout.** MAML or Meta-SGD with our learnable input-dependent noise generator.

**Datasets** We validate our method on the following two benchmark datasets for few-shot classification. **1) Omniglot:** This gray-scale hand-written character dataset consists of 1623 classes with 20 examples of size $28 \times 28$ for each class. Following the experimental setup of Vinyals et al. (2016), we use 1200 classes for meta-training, and the remaining 423 classes for meta-testing. We further augment classes by rotating 90 degrees multiple times, such that the total number of classes is $1623 \times 4$. **2) miniImageNet:** This is a subset of ILSVRC-2012 (Deng et al., 2009), consisting of 100 classes with 600 examples of size $84 \times 84$ per each class. There are 64, 16 and 20 classes for meta- train/validation/test respectively.

Table 1: **Few-shot classification performance** on conventional 4-layer convolutional neural networks. All reported results are average performances over 1000 randomly selected episodes with standard errors for 95% confidence interval over tasks.

| Models | Omniglot 20-way | | miniImageNet 5-way | |
|---|---|---|---|---|
| | 1-shot | 5-shot | 1-shot | 5-shot |
| Meta-Learning LSTM (Ravi & Larochelle, 2017) | - | - | $43.44_{\pm 0.77}$ | $60.60_{\pm 0.71}$ |
| Matching Networks (Vinyals et al., 2016) | 93.8 | 98.7 | $43.56_{\pm 0.84}$ | $55.31_{\pm 0.73}$ |
| Prototypical Networks (Snell et al., 2017) | 95.4 | 98.7 | $46.14_{\pm 0.77}$ | $65.77_{\pm 0.70}$ |
| Prototypical Networks (Snell et al., 2017) (Higher way) | 96.0 | 98.9 | $49.42_{\pm 0.78}$ | $\mathbf{68.20_{\pm 0.66}}$ |
| MAML (our reproduction) | $95.23_{\pm 0.17}$ | $98.38_{\pm 0.07}$ | $49.58_{\pm 0.65}$ | $64.55_{\pm 0.52}$ |
| Meta-SGD (our reproduction) | $96.16_{\pm 0.14}$ | $98.54_{\pm 0.07}$ | $48.30_{\pm 0.64}$ | $65.55_{\pm 0.56}$ |
| Reptile (Nichol et al., 2018) | $89.43_{\pm 0.14}$ | $97.12_{\pm 0.32}$ | $49.97_{\pm 0.32}$ | $65.99_{\pm 0.58}$ |
| Amortized Bayesian ML (Ravi & Beatson, 2019) | - | - | $45.00_{\pm 0.60}$ | - |
| Probabilistic MAML (Finn et al., 2018) | - | - | $50.13_{\pm 1.86}$ | - |
| MT-Net (Lee & Choi, 2018) | $96.2_{\pm 0.4}$ | - | $51.70_{\pm 1.84}$ | - |
| CAVIA (512) (Zintgraf et al., 2019) | - | - | $51.82_{\pm 0.65}$ | $65.85_{\pm 0.55}$ |
| **MAML + Meta-dropout** | $96.63_{\pm 0.13}$ | $98.73_{\pm 0.06}$ | $\mathbf{51.93_{\pm 0.67}}$ | $\mathbf{67.42_{\pm 0.52}}$ |
| **Meta-SGD + Meta-dropout** | $\mathbf{97.02_{\pm 0.13}}$ | $\mathbf{99.05_{\pm 0.05}}$ | $50.87_{\pm 0.63}$ | $65.55_{\pm 0.57}$ |

(a) MAML (miniImageNet)          (b) Meta-dropout (miniImageNet)          (c) Meta-dropout (Omniglot)

Figure 4: **Visualization of the task-specific decision boundaries** in the last latent feature space. We use the model trained with 1-shot. The visualizations are the projection of the features after completing the last ($5_{th}$) inner-gradient step, where the sampled 4 examples (2 examples ($\bigcirc$,$\triangle$) $\times$ 2 classes) participate in the inner-optimization. (a) and (b) are drawn from the same task. See Appendix C for the details about this visualization.

**Base networks.** We use the standard network architecture for the evaluation of few-shot classification performance (Finn et al., 2017), which consists of 4 convolutional layers with $3 \times 3$ kernels ("same" padding), and has either 64 (Omniglot) or 32 (miniImageNet) channels. Each layer is followed by batch normalization, ReLU, and max pooling ("valid" padding).

**Experimental setup.** **Omniglot:** For 1-shot classification, we use the meta-batchsize of $B = 8$ and the inner-gradient stepsize of $\alpha = 0.1$. For 5-shot classification, we use $B = 6$ and $\alpha = 0.4$. We train for total $40K$ iterations with meta-learning rate $10^{-3}$. **mimiImageNet:** We use $B = 4$ and $\alpha = 0.01$. We train for $60K$ iterations with meta-learning rate $10^{-4}$. **Both datasets:** Each inner-optimization consists of 5 SGD steps for both meta-training and meta-testing. Each task consists of 15 test examples. Note that the testing framework becomes transductive via batch normalization, as done in Finn et al. (2017). We use Adam optimizer (Kingma & Ba, 2014) with gradient clipping of $[-3, 3]$. We used TensorFlow (Abadi et al., 2016) for all our implementations.

## 4.1 FEW-SHOT CLASSIFICATION EXPERIMENTS

Table 1 shows the classification results obtained with conventional 4-layer convolutional neural networks on Omniglot and mimiImageNet dataset. The base MAML or Meta-SGD with Meta-dropout outperform all the baselines, except for the Prototypical Networks trained with "higher way" (20-way) on the miniImageNet 5-shot classification[1]. Figure 4 visualizes the learned decision boundaries of MAML and meta-dropout. We observe that the perturbations from meta-dropout generate datapoints that are close to the decision boundaries for the classification task at the test time, which could effectively improve the generalization accuracy. See Appendix section E for more visualizations, including stochastic activations and input-level projections of perturbations.

---

[1] Strictly, this result is not comparable with others as all other models are trained with 5-way classification problems during meta-training.

Table 2: **Comparison against existing perturbation-based regularization techniques.** The performances are obtained by applying the regularizers to the inner gradient steps of MAML.

| Models (MAML +) | Noise Type | Hyper-parameter | Omniglot 20-way | | miniImageNet 5-way | |
|---|---|---|---|---|---|---|
| | | | 1-shot | 5-shot | 1-shot | 5-shot |
| No perturbation | | None | $95.23_{\pm0.17}$ | $98.38_{\pm0.07}$ | $49.58_{\pm0.65}$ | $64.55_{\pm0.52}$ |
| Input & Manifold Mixup | | $\gamma = 0.2$ | $89.78_{\pm0.25}$ | $97.86_{\pm0.08}$ | $48.62_{\pm0.66}$ | $63.86_{\pm0.53}$ |
| (Zhang et al., 2017) | Pairwise | $\gamma = 1$ | $87.00_{\pm0.28}$ | $97.27_{\pm0.10}$ | $48.24_{\pm0.62}$ | $62.32_{\pm0.54}$ |
| (Verma et al., 2019) | | $\gamma = 2$ | $87.26_{\pm0.28}$ | $97.14_{\pm0.17}$ | $48.42_{\pm0.64}$ | $62.56_{\pm0.55}$ |
| Variational | | $\beta = 10^{-5}$ | $92.09_{\pm0.22}$ | $98.85_{\pm0.07}$ | $48.12_{\pm0.65}$ | $64.78_{\pm0.54}$ |
| Information Bottleneck | Add. | $\beta = 10^{-4}$ | $93.01_{\pm0.20}$ | $98.80_{\pm0.07}$ | $46.75_{\pm0.63}$ | $64.07_{\pm0.54}$ |
| (Alemi et al., 2017) | | $\beta = 10^{-3}$ | $94.98_{\pm0.16}$ | $98.75_{\pm0.07}$ | $47.59_{\pm0.60}$ | $63.30_{\pm0.53}$ |
| Information Dropout | | $\beta = 10^{-5}$ | $94.49_{\pm0.17}$ | $98.50_{\pm0.07}$ | $50.36_{\pm0.68}$ | $65.91_{\pm0.55}$ |
| (ReLU ver.) | Mult. | $\beta = 10^{-4}$ | $94.36_{\pm0.17}$ | $98.53_{\pm0.07}$ | $49.14_{\pm0.63}$ | $64.96_{\pm0.54}$ |
| (Achille & Soatto, 2018) | | $\beta = 10^{-3}$ | $94.28_{\pm0.17}$ | $98.65_{\pm0.07}$ | $43.78_{\pm0.61}$ | $63.36_{\pm0.56}$ |
| **Meta-dropout** | Add. | 0.1 | $96.55_{\pm0.14}$ | $\mathbf{99.04_{\pm0.05}}$ | $50.25_{\pm0.66}$ | $66.78_{\pm0.53}$ |
| (See Appendix B for Add.) | **Mult.** | None | $\mathbf{96.63_{\pm0.13}}$ | $98.73_{\pm0.06}$ | $\mathbf{51.93_{\pm0.67}}$ | $\mathbf{67.42_{\pm0.52}}$ |

**Comparison against perturbation-based regularization methods**  In Table 2, we compare with the existing regularization methods based on input-dependent perturbation, such as Mixup (Zhang et al., 2017; Verma et al., 2019) and Information-theoretic regularizers (Achille & Soatto, 2018; Alemi et al., 2017). We train the base MAML with the baseline regularizers added to the inner-gradient steps, and set the number of perturbations for each step to 1 for meta-training and 30 for meta-testing, as in the case of Meta-dropout. We meta-train additional parameters from the baselines by optimizing them in the inner-gradient steps for fair comparison.

We first compare meta-dropout against mixup variants. The perturbation for mixup regularization is defined as a linear interpolation of both inputs and labels between two randomly sampled training datapoints. We interpolate at both the input and manifold level following Verma et al. (2019). The interpolation ratio $\lambda \in [0, 1]$ follows $\lambda \sim \text{Beta}(\gamma, \gamma)$, and we use $\gamma \in \{0.2, 1, 2\}$ following the settings of the original paper. Table 2 shows that Mixup regularization significantly degrades the few-shot classification performance within the range of $\gamma$ we considered. This is because, in the meta-learning framework, the interpolations of each task-adaptation process ignores the larger task distribution, which could conflict with the previously accumulated meta-knowledge.

Next, we compare with the information-theoretic regularizers. Information Bottleneck (IB) method (Tishby & Zaslavsky, 2015) is a framework for searching for the optimal tradeoff between forgetting of unnecessary information in inputs (nuisances) and preservation of information for correct prediction, with the hyperparameter $\beta$ controlling the tradeoff. The higher the $\beta$, the more strongly the model will forget the inputs. We consider the two recent variations for IB-regularization for deep neural networks, namely Information Dropout (Achille & Soatto, 2018) and Variational Information Bottleneck (VIB) (Alemi et al., 2017). Those information-theoretic regularizers are relevant to meta-dropout as they also inject input-dependent noise, which could be either multiplicative (Information Dropout) or additive (VIB). However, the assumption of optimal input forgetting does not hold in meta-learning framework, because it will forget the previous task information as well. Table 2 shows that these regularizers significantly underperform ours in the meta-learning setting.

**Adversarial robustness**  Lastly, we investigate if the perturbations generated from meta-dropout can improve adversarial robustness (Goodfellow et al., 2015) as well. Projected gradient descent (PGD) (Madry et al., 2017) attack of $\ell_1$, $\ell_2$, and $\ell_\infty$ norm is considered. As an adversarial learning baseline on MAML framework, we compare with a variant of MAML that adversarially perturbs each training example in its inner-gradient steps (10 PGD steps with radius $\epsilon$ for each inner-gradient step[2]). Their outer objective maintains as test loss for fair comparison with other models. At meta-test time, we test all the baselines with PGD attack (200 steps) of all types of perturbations.

In Figure 5, we see that whereas the baseline regularizers such as manifold-mixup and information-theoretic regularizers[3] fail to defend against adversarial attacks, meta-dropout not only improves

---

[2] We choose $\epsilon$ as minimum and maximum value within the overall attack range for the test examples.

[3] For the hyperparameters of each baseline, we choose the one with the best accuracy in Table 2. We verified that for those baselines, interestingly, adversarial robustness is not much sensitive to hyperparameter selection.

Table 3: **Ablation study on the noise types** applied to the inner-gradient steps. (✓) means that the baseline is a core component of Meta-dropout.)

| Models (MAML+) | Rand. samp. | Learn. mult. | Input dep. | Omniglot 20-way | | miniImageNet 5-way | |
|---|---|---|---|---|---|---|---|
| | | | | 1-shot | 5-shot | 1-shot | 5-shot |
| None | X | X | X | $95.23_{\pm0.17}$ | $98.38_{\pm0.07}$ | $49.58_{\pm0.65}$ | $64.55_{\pm0.52}$ |
| Fixed Gaussian (✓) | O | X | X | $95.44_{\pm0.17}$ | $\mathbf{98.99_{\pm0.06}}$ | $49.39_{\pm0.63}$ | $66.84_{\pm0.54}$ |
| Weight Gaussian | O | X | X | $94.32_{\pm0.18}$ | $98.35_{\pm0.07}$ | $49.37_{\pm0.64}$ | $64.78_{\pm0.54}$ |
| Independent Gaussian | O | O | X | $94.36_{\pm0.18}$ | $98.26_{\pm0.08}$ | $50.31_{\pm0.64}$ | $66.97_{\pm0.54}$ |
| MAML + More param | X | O | O | $95.83_{\pm0.15}$ | $97.85_{\pm0.09}$ | $50.63_{\pm0.64}$ | $65.20_{\pm0.51}$ |
| Determ. Meta-drop. (✓) | X | O | O | $95.99_{\pm0.14}$ | $97.78_{\pm0.09}$ | $50.75_{\pm0.63}$ | $65.62_{\pm0.53}$ |
| Meta-drop. w/ learned var. | O | O | O | $95.98_{\pm0.15}$ | $98.87_{\pm0.06}$ | $50.93_{\pm0.68}$ | $66.15_{\pm0.56}$ |
| **Meta-dropout** | O | O | O | $\mathbf{96.63_{\pm0.13}}$ | $98.73_{\pm0.06}$ | $\mathbf{51.93_{\pm0.67}}$ | $\mathbf{67.42_{\pm0.52}}$ |

(a) Omniglot 1-shot ($\ell_1$)  (b) Omniglot 1-shot ($\ell_2$)  (c) Omniglot 1-shot ($\ell_\infty$)  (d) Omniglot 5-shot ($\ell_\infty$)

Figure 5: **Adversarial robustness** against PGD attack with varying size of radius $\epsilon$. The region of clean accuracies are magnified for better visualization.

generalization (clean accuracies), but also improves adversarial robustness significantly. It means that the perturbation directions toward test examples can generalize to adversarial perturbations. This result is impressive considering neither the inner nor the outer optimization of meta-dropout involves explicit adversarial training. Although we need further research to clarify the relationship between the two, the results are exciting since existing models for adversarial learning achieve robustness at the expense of generalization accuracy. See Stutz et al. (2019) for the discussion about disentangling between generalization and adversarial robustness. The results from miniImageNet dataset is in Appendix D. Although the gap is not as significant as in Omniglot dataset, the overall tendencies look similar.

Moreover, Figure 6(a), 6(b) and 6(c) shows that meta-dropout can even generalize across *multiple types of adversarial attacks* we considered. Generalization between multiple types of perturbations is only recently investigated by Tramèr & Boneh (2019), which is a very important problem as most of the existing adversarial training methods target a single perturbation type at a time, limiting their usefulness. We believe that the results in Figure 5 can have a great impact in this research direction.

**Ablation study** In Table 3, we compare our model against other types of noise generator, to justify the use of input-dependent multiplicative noise for meta-dropout. We describe the baseline noise generators as follows: **1) Fixed Gaussian:** MAML with fixed multiplicative noise, such that $\mathbf{a}$ in Eq. 5 follows $\mathbf{a} \sim \mathcal{N}(\mathbf{0}, \mathbf{I})$ without any trainable parameters. **2) Weight Gaussian:** MAML with fully factorized Gaussian weights whose variances are meta-learned. **3) Independent Gaussian:** MAML with input-independent multiplicative noise, such that $\mathbf{a} \sim \mathcal{N}(\boldsymbol{\mu}, \mathbf{I})$, where learnable parameter $\boldsymbol{\mu}$ has the same dimensionality to the channels (similarly to channel-wise dropout). We see from the Table 3 that neither the fixed distribution (Fixed Gaussian) nor the input-independent noise (Weight Gaussian and Independent) outperforms the base MAML, whereas our meta-dropout model significantly outperforms MAML. The results support our claim that the input-dependent noise generators are more transferable across tasks, as it can generate noise distribution adaptively for each instance.

We also compare with the MAML with additional parameters $\phi$ (deterministic $\mathbf{a}$ in Eq. 5) that participate in the inner-gradient steps as well, which we denote as (MAML + More param). We get slight improvements from the base MAML by doing so. Deterministic Meta-dropout is MAML + More param with $\phi$ *not* participating in the inner-gradient steps as with meta-dropout. Although there are some gain from deterministically meta-learning the representation space with $\phi$, we can significantly improve the performance by injecting stochastic noise (meta-dropout). Lastly, we com-

pare against a variant of meta-dropout that instead of using fixed $\mathbf{I}$ in Eq. 5, learns the variance in an input-dependent manner (meta-dropout w/ learned variance) but it underperforms meta-dropout with fixed variance.

Note that the slightly good performance of Deterministic Meta-dropout and Fixed Gaussian does not hurt our claim on the effectiveness of multiplicative noise, because multiplicative noise, by definition, consists of two parts: deterministic multiplication and pure random noise. For example, Bernoulli dropout consists of Bernoulli retain probability $0 \leq p \leq 1$ (deterministic multiplication) and actual random sampling (pure random noise). In this vein, we can say that Deterministic Meta-dropout demonstrates the effectiveness of meta-learning the probability $p$, and Fixed Gaussian shows the effectiveness of injecting (multiplying) pure random noise $\mathcal{N}(\mathbf{0}, \mathbf{I})$ on each feature location. Overall, our meta-dropout combines the two components, which are complementary each other, into a novel input-dependent form of perturbation function.

## 5 CONCLUSION

We proposed a novel regularization method for deep neural networks, *meta-dropout*, which *learns to perturb* the latent features of training examples for improved generalization. However, learning how to optimally perturb the input is difficult in a standard learning scenario, as we do not know which data will be given at the test time. Thus, we tackle this challenge by learning the input-dependent noise generator in a meta-learning framework, to explicitly train it to minimize the test loss during meta-training. Our noise learning process could be interpreted as meta-learning of a variational inference for a specific graphical model in Fig. 3. We validate our method on benchmark datasets for few-shot classification, on which it significantly improves the generalization performance of the target meta-learning model, while largely outperforming existing regularizers based on input-dependent perturbation. As future work, we plan to investigate the relationships between generalization and adversarial robustness.

**Acknowledgements** This work was supported by Google AI Focused Research Award, the Engineering Research Center Program through the National Research Foundation of Korea (NRF) funded by the Korean Government MSIT (NRF-2018R1A5A1059921), Institute of Information & communications Technology Planning & Evaluation (IITP) grant funded by the Korea government (MSIT) (No.2019-0-00075), and Artificial Intelligence Graduate School Program (KAIST)).

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

## A   ALGORITHM

We provide the pseudocode for meta-training and meta-testing of Meta-dropout.

---

**Algorithm 1** Meta-training

1: **Input:** Task distribution $p(\mathcal{T})$, Number of inner steps $K$, Inner step size $\alpha$, Outer step size $\beta$.
2: **while** not converged **do**
3:     Sample $(\mathcal{D}^{\text{tr}}, \mathcal{D}^{\text{te}}) \sim p(\mathcal{T})$
4:     $\boldsymbol{\theta}_0 \leftarrow \boldsymbol{\theta}$
5:     **for** $k = 0$ to $K - 1$ **do**
6:         Sample $\tilde{\mathbf{z}}_i \sim p(\mathbf{z}_i | \mathbf{x}_i^{\text{tr}}; \boldsymbol{\phi}, \boldsymbol{\theta}_k)$   for   $i = 1, \dots, N$
7:         $\boldsymbol{\theta}_{k+1} \leftarrow \boldsymbol{\theta}_k + \alpha \nabla_{\boldsymbol{\theta}_k} \frac{1}{N} \sum_{i=1}^{N} \log p(\mathbf{y}_i^{\text{tr}} | \mathbf{x}_i^{\text{tr}}, \tilde{\mathbf{z}}_i; \boldsymbol{\theta}_k)$
8:     **end for**
9:     $\boldsymbol{\theta}^* \leftarrow \boldsymbol{\theta}_K$
10:     $\boldsymbol{\theta} \leftarrow \boldsymbol{\theta} + \beta \frac{1}{M} \sum_{j=1}^{M} \nabla_{\boldsymbol{\theta}} \log p(\mathbf{y}_j^{\text{te}} | \mathbf{x}_j^{\text{te}}, \mathbf{z}_j = \bar{\mathbf{z}}_j; \boldsymbol{\theta}^*)$
11:     $\boldsymbol{\phi} \leftarrow \boldsymbol{\phi} + \beta \frac{1}{M} \sum_{j=1}^{M} \nabla_{\boldsymbol{\phi}} \log p(\mathbf{y}_j^{\text{te}} | \mathbf{x}_j^{\text{te}}, \mathbf{z}_j = \bar{\mathbf{z}}_j; \boldsymbol{\theta}^*)$
12: **end while**

---

**Algorithm 2** Meta-testing

1: **Input:** Number of inner steps $K$, Inner step size $\alpha$, MC sample size $S$.
2: **Input:** Learned parameter $\boldsymbol{\theta}$ and $\boldsymbol{\phi}$ from Algorithm 1.
3: **Input:** Meta-test dataset $(\mathcal{D}^{\text{tr}}, \mathcal{D}^{\text{te}})$.
4: $\boldsymbol{\theta}_0 \leftarrow \boldsymbol{\theta}$
5: **for** $k = 0$ to $K - 1$ **do**
6:     Sample $\left\{ \mathbf{z}_i^{(s)} \right\}_{s=1}^{S} \overset{\text{i.i.d}}{\sim} p(\mathbf{z}_i | \mathbf{x}_i^{\text{tr}}; \boldsymbol{\phi}, \boldsymbol{\theta}_k)$   for   $i = 1, \dots, N$
7:     $\boldsymbol{\theta}_{k+1} \leftarrow \boldsymbol{\theta}_k + \alpha \nabla_{\boldsymbol{\theta}_k} \frac{1}{N} \sum_{i=1}^{N} \frac{1}{S} \sum_{s=1}^{S} \log p(\mathbf{y}_i^{\text{tr}} | \mathbf{x}_i^{\text{tr}}, \mathbf{z}_i^{(s)}; \boldsymbol{\theta}_k)$
8: **end for**
9: $\boldsymbol{\theta}^* \leftarrow \boldsymbol{\theta}_K$
10: Evaluate $p(\mathbf{y}_j^{\text{te}} | \mathbf{x}_j^{\text{te}}, \mathbf{z}_j = \bar{\mathbf{z}}_j; \boldsymbol{\theta}^*)$   for   $j = 1, \dots, M$

---

## B   META-DROPOUT WITH ADDITIVE NOISE

In this section, we introduce the additive noise version of our meta-dropout, introduced in Table 2. At each layer $l$, we add Gaussian noise $\mathbf{z}^{(l)}$ whose mean is zero and covariance is diagonal, i.e. $\mathbf{z}^{(l)} \sim \mathcal{N}(\mathbf{z}^{(l)} | \mathbf{0}, \text{diag}(\boldsymbol{\sigma}^2))$, to the preactivation features at each layer, i.e. $\mathbf{f}^{(l)} := \mathbf{f}^{(l)}(\mathbf{h}^{(l-1)})$. The diagonal covariance is generated from each previous layer's activation $\mathbf{h}^{(l-1)}$ in input-dependent manner. i.e. $\boldsymbol{\sigma}^{(l)} := \boldsymbol{\sigma}(\mathbf{h}^{(l-1)})$, and we meta-learn such covariance networks $\boldsymbol{\sigma}^{(1)}, \dots, \boldsymbol{\sigma}^{(L)}$ over the task distribution. Putting together,

$$\mathbf{z}^{(l)} \sim \mathcal{N}(\mathbf{z}^{(l)} | \mathbf{0}, \lambda^2 \, \text{diag}(\boldsymbol{\sigma}^2)), \tag{9}$$

$$\mathbf{h}^{(l)} = \text{ReLU}(\mathbf{f}^{(l)} + \mathbf{z}^{(l)}). \tag{10}$$

where $\lambda$ is a hyperparameter for controlling how far each noise variable can spread out to cover nearby test examples. We fix $\lambda = 0.1$ in all our experiments, that seems to help stabilize the overall training process. Reparameterization trick is applied to obtain stable and unbiased gradient estimation w.r.t. the mean and variance of the Gaussian distribution, following Kingma and Welling Kingma & Welling (2014).

We empirically found that this additive noise version of meta-dropout performs well on Omniglot dataset, but shows somewhat suboptimal performance on more complicated dataset such as miniImageNet (See Table 2). We may need more research on different noise types, but for now, we suggest to use the multiplicative version of meta-dropout in Eq. 5 which seems to perform well on most of the cases.

## C  HOW TO VISUALIZE DECISION BOUNDARY

Visualization in Figure 4 shows 2D-mapped latent features and binary classification decision boundary of two random classes in a task. After parameters are adapted to the given task, we make binary classifier of random classes $c_1$ and $c_2$ by taking only two columns of final linear layer parameters : $\mathbf{W} = [\mathbf{w}_{c_1}, \mathbf{w}_{c_2}]$ and $\mathbf{b} = [b_{c_1}, b_{c_2}]$. Then we compute the last latent features $\mathbf{H}^{(L)}$ of data points in the classes.

The decision hyperplane of the linear classifier in the latent space is given as,

$$\mathbf{h}_{db}^{\top}(\mathbf{w}_{c_1} - \mathbf{w}_{c_2}) + (b_{c_1} - b_{c_2}) = 0 \tag{11}$$

X-coordinates in our 2D visualization are inner product values between latent features and the normal vector of the decision hyperplane i.e. latent features are projected to orthogonal direction of the decision hyperplane.

$$\mathbf{c}^x = \mathbf{H}^{(L)} \frac{\mathbf{w}_{c_1} - \mathbf{w}_{c_2}}{||\mathbf{w}_{c_1} - \mathbf{w}_{c_2}||} \tag{12}$$

Y-coordinates are determined by t-distributed stochastic neighbor emgedding (t-SNE) to reduce all other dimensions.

$$\mathbf{c}^y = tSNE \left( \mathbf{H}^{(L)} - \mathbf{c}_x \frac{(\mathbf{w}_{c_1} - \mathbf{w}_{c_2})^{\top}}{||\mathbf{w}_{c_1} - \mathbf{w}_{c_2}||} \right) \tag{13}$$

The points on the decision hyperplane is projected to a vertical line with following x-coordinate in the 2D space.

$$c_{db}^x = \mathbf{h}_{db}^{\top} \frac{\mathbf{w}_{c_1} - \mathbf{w}_{c_2}}{||\mathbf{w}_{c_1} - \mathbf{w}_{c_2}||} = -\frac{b_{c_1} - b_{c_2}}{||\mathbf{w}_{c_1} - \mathbf{w}_{c_2}||} \tag{14}$$

## D  ADVERSARIAL ROBUSTNESS EXPERIMENT WITH MINIIMAGENET

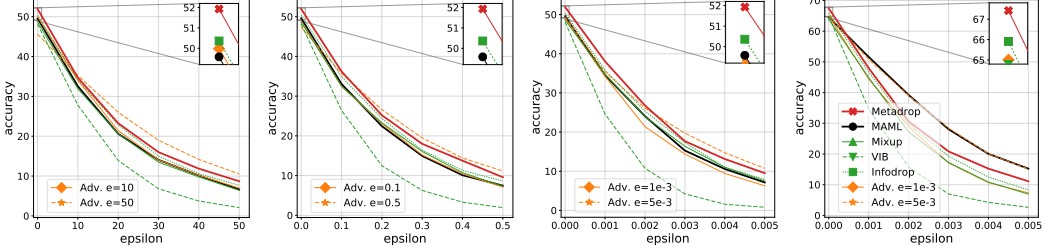

(a) m.ImgNet 1-shot ($\ell_1$)  (b) m.ImgNet 1-shot ($\ell_2$)  (c) m.ImgNet 1-shot ($\ell_\infty$)  (d) m.ImgNet 5-shot ($\ell_\infty$)

Figure 6: **Adversarial robustness** against PGD attack with varying size of radius $\epsilon$. The region of clean accuracies are magnified for better visualization.

Figure 6 plots the clean and adversarial accuracies over varying size of $\epsilon$. Although the difference between models are not as remarkable as in Figure 5, our meta-dropout seems to be the only model that improves both clean and adverarial accuracies. More research seems required to investigate what makes difference between Omniglot and mimiImageNet dataset.

## E  MORE VISUALIZATIONS

In order to see the meaning of perturbation at input level, we reconstruct expected perturbed image from perturbed features. A separate deconvolutional decoder network is trained to reconstruct original image from unperturbed latent features, then the decoder is used to reconstruct perturbed image from perturbed features of 3rd layer.

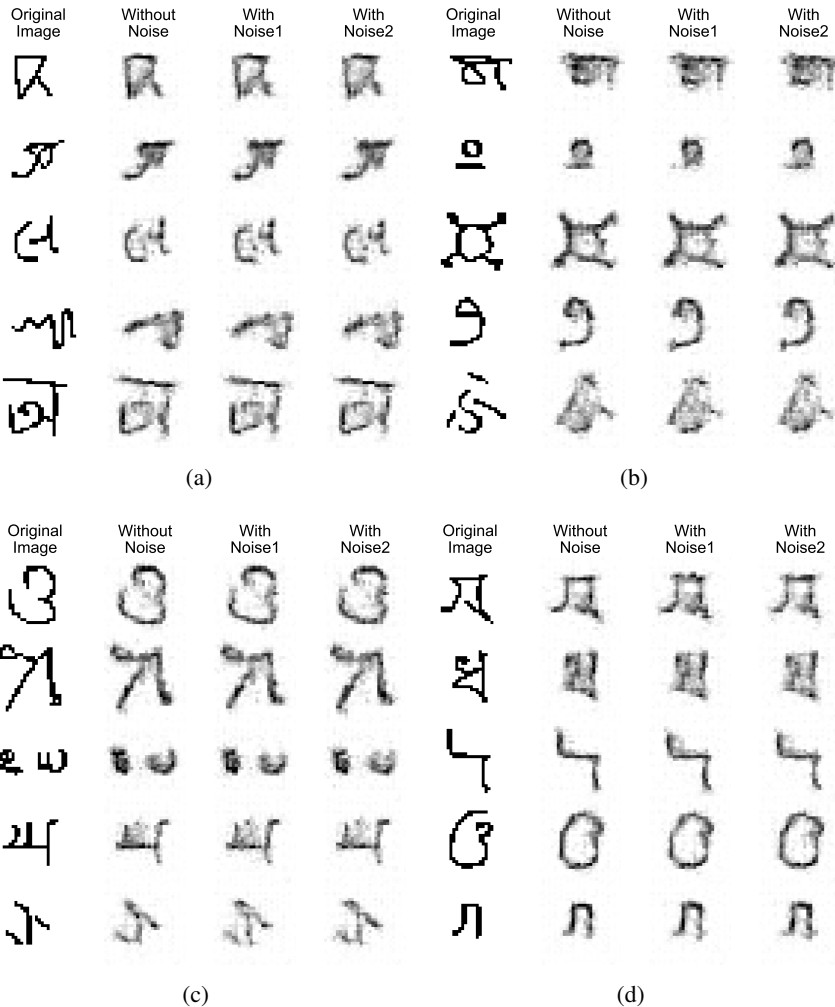

Figure 7: **Feature Visualization** Layer activations and perturbed activations in Omniglot datset.

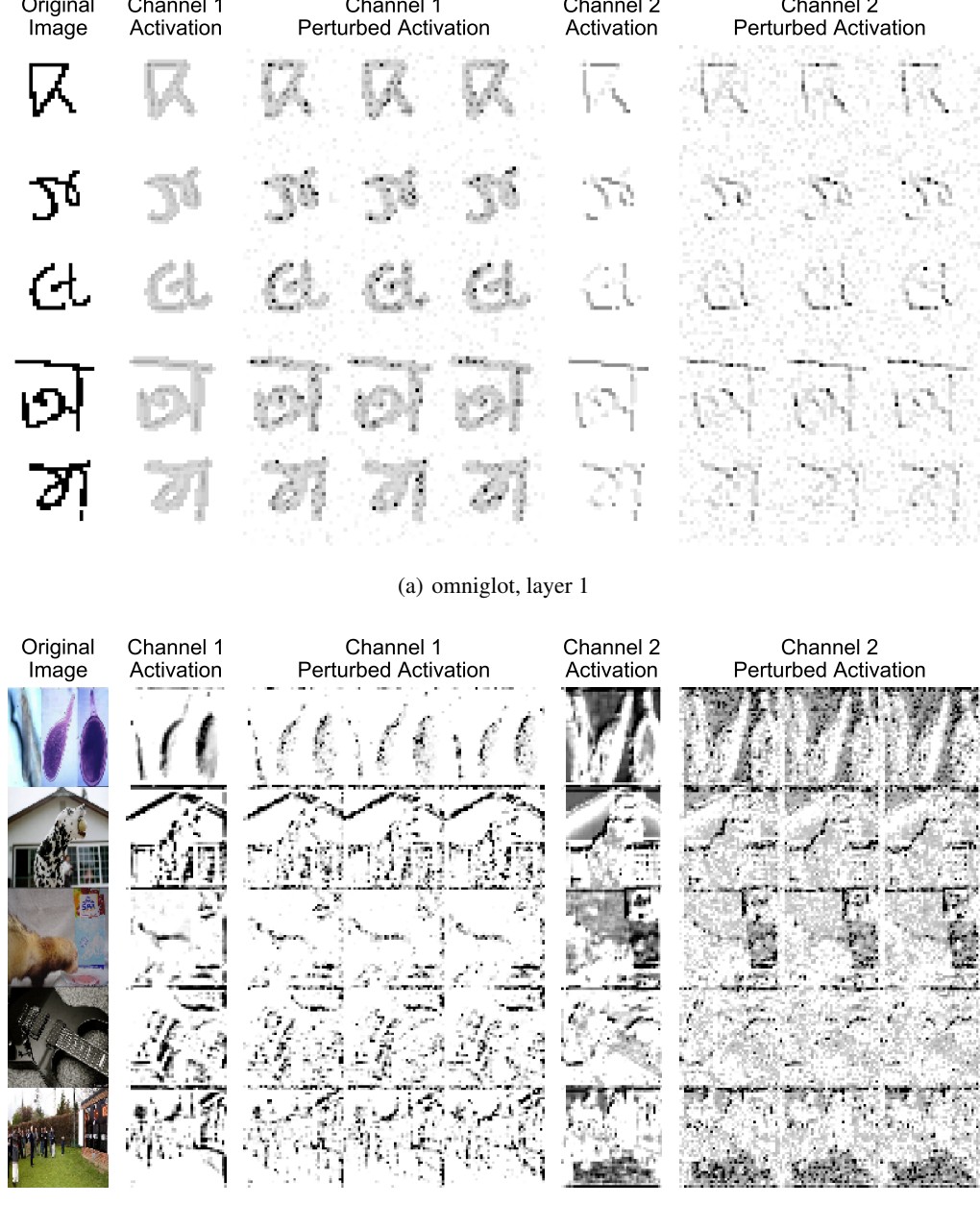

(a) omniglot, layer 1

(b) mimiImageNet, layer 2

Figure 8: **Visualization of stochastic features.** We visualize the deterministic activations and the corresponding stochastic activations with perturbation.

