# OpenReview forum: "Meta Dropout: Learning to Perturb Latent Features for Generalization"
_ICLR.cc/2020/Conference — Accept (Poster)_

### Official Review · AnonReviewer3 · 2019-10-23
**Official Blind Review #3**

**Rating:** 3

**Review:**

The paper proposes learning to add input-dependent noise to improve the generalization of MAML-style meta-learning algorithm. The proposed method is evaluated on OmniGlot and miniImageNet. The paper reports improvements upon MAML, MAML with meta-learned parameter-wise learning rates, as well as a few regularization methods that are based on input/hidden state perturbations (Mixup, Variational Information Bottleneck). An ablation study also compares the proposed meta-dropout algorithm with a number of modifications, such as a fixed noise, input-independent noise, etc. It is furthermore shown that meta-dropout somewhat improves the model’s robustness against an adversarial attack.

The paper is somewhat incremental considering that Li et al, (2017) and Balaji et al, (2018) have already proposed meta-learning parameter-wise learning rates and parameter-wise regularization coefficient respectively. One difference from the methods above is that in the proposed method noise is controlled by the input. The ablation however shows that in 5-shot classification case simply adding non-trainable noise works quite well.

It seems like the choice of the particular method for adding the noise was performed using the test set. If it’s true, this is methodologically wrong: model selection should be performed on a development set (or meta-development) set. Futhermore, Table 2 contains some results named “Add.”, which I guess stands for additive noise. I did not find an explanation of what is the specific method for adding noise used in this case. Such additive noise is also missing from ablation experiments.

Overall, it seems that paper falls short of clearly proving that back-propagating through MAML to the noise parameters is helpful. The “Deterministic Meta-Dropout” performs better than baseline MAML, and arguably, meaning that some part of the improvement upon MAML can be due to the architectural differences and not due to noise. “Independent Gaussian” and “Weight Gaussian” baselines perform worse than non-trainable noise (“Fixed Gaussian”). Learning the variance for the noise is shown to be detrimental. There is just too much confusion in the results, the improvements are not very robust.

The paper writing is okay, but there are serious issues. I am not sure I understand the argument in Section 3.2 that meta-dropout performs variational inference. It seems like Equation 7 is wrong because  y_i is missing from the second argument of the KL divergence term. The transition to Equation 8 is therefore also wrong, and as far as I can understand, the whole argument breaks down. Line 7 in Algorithm 1 in Appendix A (which by the way should really be in the main text) does not make sense.

Other issues:
- the second sentence of the abstract is not implied by the first, the usage of “thus” does not seem appropriate
- the intro should probably mention L1 and L2 regularization as well
- in Section 3.1 there is a forward reference to Equation 5, makes understanding the text quite hard
- “meta-droput”, “robustenss”: typos in many places
- Figure 4 visualization is not clear.
- the architectural change required to add noise is not explained in the paper (i.e. what is \phi and how it’s used)
- no comparison to meta-learned L1 regularization
- a baseline is missing in which \phi is treated as a part of \theta and trained with vanilla MAML


**Experience Assessment:**

I have read many papers in this area.

**Review Assessment: Checking Correctness Of Derivations And Theory:**

N/A

**Review Assessment: Checking Correctness Of Experiments:**

I carefully checked the experiments.

**Review Assessment: Thoroughness In Paper Reading:**

N/A

---

> ### Author Response · Authors · 2019-11-11
> **Response to Reviewer #3 (2/2)**
>
> 5. It seems like Equation 7 is wrong because y_i is missing from the second argument of the KL divergence term. The transition to Equation 8 is therefore also wrong, and as far as I can understand, the whole argument breaks down.
>
> - This is a critical misunderstanding. Equation 7 is correct. In Equation 7, the second argument p(z|x) is called a “conditional prior”, which does not observe the label y since p(z|x) is part of the generative process. See the Equation 4 of Sohn et al. [1] which is exactly the same as the Equation 7 of our paper.
>
> It seems that you are confused with the following decomposition of log evidence:
>
> Log evidence = ELBO + $KL[q(z|x,y)\|p(z|x,y)]$,		        (a)
> ELBO = E[Log-likelihood] - $KL[q(z|x,y)\|p(z|x)]$		(b)
>
> The log evidence term is decomposed of ELBO and the KL term involving $p(z|x,y)$ in Eq. (a). However, what Eq.7 in the main paper describes is the actual ELBO expression in Eq.(b), which contains the KL divergence between the approximate posterior and the prior $p(z|x)$.
>
> Reference: [1] Sohn et al., Learning Structured Output Representation using Deep Conditional Generative Models, In NIPS, 2015.
>
>
> 6. Line 7 in Algorithm 1 in Appendix A (which by the way should really be in the main text) does not make sense. The second sentence of the abstract is not implied by the first, the usage of “thus” does not seem appropriate. The intro should probably mention L1 and L2 regularization as well. “meta-droput”, “robustenss”: typos in many places.
>
> - Thank you for pointing them out. We updated the revision based on your suggestions.
>
>
> 7. Figure 4 visualization is not clear.
>
> - Figure 4 visualizes the perturbations generated by our Meta-Dropout and the decision boundary for classification. Could you provide us more specific comments on which part of Figure 4 is not clear?
>
>
> 8. The architectural change required to add noise is not explained in the paper (i.e. what is \phi and how it’s used).
>
> - The architectural change and \phi are clearly explained in Figure 2 and the paragraph “form of the noise” at the end of page 4.
>
>
> 9. Additional baselines
>
> - We conduct experiments on additional baselines as requested:
> (1) MetaReg: MAML + L1 regularization
> (2) MAML (R3): \phi treated as \theta, as R3 requested
>
> Models				Omni-1shot	Omni-5shot	mimg-1shot	mimg-5shot
> MAML				95.23+-0.17	98.38+-0.07	49.58+-0.65	64.55+-0.52
> MetaReg (Ours)		95.28+-0.15	98.85+-0.06	49.76+-0.67	65.42+-0.53
> MAML (R3)			96.15+-0.15	98.69+-0.07	42.08+-0.63	62.82+-0.53
> Meta-dropout		96.63+-0.13	98.73+-0.06	51.93+-0.67	67.42+-0.52
>
> As shown, all the suggested baseline models largely underperform Meta-Dropout.

---

> > ### Comment · AnonReviewer3 · 2019-11-15
> > **still not convinced**
> >
> > I would like to thank the authors for the detailed response.
> >
> > It’s indeed true that the formulas in Section 3.2 are correct, I apologize for the confusion. But I agree with other reviewers that the connection between the proposed method and variational inference is not significant. There is no variational approximation in the proposed method, only the Jensen lower bound. Equation 7 could be safely skipped.
> >
> > I spent significant time deliberating whether the baseline results provided in the paper and in the rebuttal provide enough evidence to say that meta-learning input-dependent noisy regularization is useful. Let me share my thoughts.
> >
> > The proposed method can be seen as introducing 3 changes to the model and the learning algorithm:
> >
> > The first change to the model: element-wise multipliers z^(l), as well as the parameters \phi that are used to compute z^(l)
> > The second change to the model: adding a form of multiplicative noise
> > A change to the algorithm: the parameters \phi are now treated differently and receive a different kind of derivative
> >
> > Meaningful subsets of the three proposed changes include:
> > (1), whereby \phi and \theta are treated identically and no noise is used. I am not sure if this is what the authors tried in their rebuttal, they did not say if they kept noise on. This is an important baseline because it isolates the change to the model.
> > (1, 2), which is to my best understanding what the authors did in the rebuttal when they tried “\phi treated as \theta”
> > (2): “Fixed Gaussian” in Table 3
> > (1, 3): “Deterministic Meta-Dropout” in Table 3
> > (2, 3): “Independent Gaussian” in Table 3
> >
> > To justify that all these 3 changes are needed, one needs to show in several experiments how the combination of all 3 performs better than subsets. The paper performs 4 experiments: 1-shot Omniglot, 5-shot Omniglot, 1-shot MiniImageNet, 5-shot MiniImageNet.
> >
> > Here’s my summary of the results:
> > 1-shot OmniGlot: the proposed method works significantly better than baselines
> > 5-shot OmniGlot: using just (2) performs as well as the proposed method
> > 1-shot ImageNet: the proposed method works significantly better than baselines, but the 42.08 accuracy that is reported in the rebuttal for (1, 2) baseline looks not very trust-worthy. Why would there be such a deterioration in this case, when in OmniGlot 1-shot case this baseline almost approached Meta-Dropout?
> > 5-shot ImageNet: the 95% confidence intervals overlap for (1,3), (2,3) and the proposed method
> > Additionally, to my best understanding, the baseline (1) was not tried. That said, I still can’t be sure that the main source of improvement is not the change in the model.
> >
> > Given this summary of the results, I am not convinced that the paper clearly shows the benefit of meta-learning input-dependent noisy regularization.
> >
> > Lastly, I disagree with the authors that the paper is clear w.r.t. what exactly \phi is and how exactly is used. I did not find this information in Figure 2 and “Form of the noise” paragraph.

---

> ### Author Response · Authors · 2019-11-11
> **Response to Reviewer #3 (1/2)**
>
> We appreciate your constructive comments. We respond to each comment as follows.
>
> 1. The paper is somewhat incremental considering that Li et al, (2017) and Balaji et al., (2018) have already proposed meta-learning parameter-wise learning rates and parameter-wise regularization coefficient respectively.
>
> - This is a critical misunderstanding. The two papers you mentioned are not even superficially similar to our model. First Meta-SGD (Li et al., 2017) aims to meta-learn the element-wise learning rate, which is completely orthogonal to our model that meta-learns input-dependent perturbation of latent features. It also largely underperforms our method (See Table 1).
>
> Meta-Reg (Balaji et al., 2018) meta-learns the hyperparameter for a global regularizer while our Meta-Dropout directly learns to perturb each feature in an input-dependent manner. and aims to tackle a completely different problem of domain generalization. Nonetheless, to the best of our efforts, we tried to import the MetaReg idea to the conventional few-shot classification setting, such that for each task, its inner-optimization objective includes L1 regularization whose coefficients are element-wisely meta-learned. The results are as follows.
>
> Models		        	Omni-1shot	Omni-5shot	mimg-1shot	mimg-5shot
> MAML		        	95.23+-0.17	98.38+-0.07	49.58+-0.65	64.55+-0.52
> MetaReg (Ours)		95.28+-0.15	98.85+-0.06	49.76+-0.67	65.42+-0.53
> Meta-dropout		96.63+-0.13	98.73+-0.06	51.93+-0.67	67.42+-0.52
>
> As shown, Meta-dropout largely outperforms MetaReg, demonstrating the effectiveness of our framework, which meta-learns the input-dependent perturbation function.
>
>
> 2. It seems like the choice of the particular method for adding the noise was performed using the test set.
>
> - This is a complete misunderstanding. We never used a meta-test set for training or hyperparameter tuning. Test in the paper refers to the instances *simulating* test instances within the meta-training dataset, which is a common terminology in meta-learning.
>
>
> 3. Table 2 contains some results named “Add.”, which I guess stands for additive noise. I did not find an explanation of what is the specific method for adding noise used in this case.
>
> - We apologize for the confusion. This is indeed an additive noise version of our meta-dropout. We updated the revision with the full description of the method in Appendix B. Since we compare against this additive version in the main table (Table 2), which is already an ablation study of the two, it is excluded from the Ablation study in Table 3.
>
>
> 4. Overall, it seems that paper falls short of clearly proving that back-propagating through MAML to the noise parameters is helpful.
>
> - The slightly good performance of “Deterministic Meta-dropout” and “Fixed Gaussian” does not hurt our claim on the effectiveness of multiplicative noise, because multiplicative noise, by definition, consists of two parts: deterministic multiplication and pure random noise. For example, Bernoulli dropout consists of Bernoulli retain probability $0 \leq p \leq 1$ (deterministic multiplication) and actual random sampling (pure random noise). In this vein, we can say that “Deterministic Meta-dropout “demonstrates the effectiveness of meta-learning the probability p, and “Fixed Gaussian” shows the effectiveness of injecting (multiplying) pure random noise N(0,I) on each feature location.
>
> Overall, our meta-dropout combines the two components, which are complementary each other, into a novel input-dependent form of perturbation function. In this regard, we have clearly demonstrated that meta-learning input-dependent multiplicative noise is beneficial for improving generalization, jointly as well as component-wisely.

---

### Official Review · AnonReviewer2 · 2019-10-24
**Official Blind Review #2**

**Rating:** 8

**Review:**

The authors propose to meta-learn, using MAML, the mean of an elementwise, input-dependent, multiplicative noise to improve generalization in few-shot learning.
The motivation is that meta-learning the noise allows to learn how to best perturb examples in order to improve generlization.  This claim is supported by ample experimental evidence and comparisons against many baselines, as well as additional ablation studies w.r.t design choices of the algorithm itself. The paper is well written and easy to read. Consequently, I think this is a nice paper and should be accepted.

Edit (leaving everything else unchanged for now): After reading R3's assessment, I agree with them that it's worrying that the Deterministic Meta-Dropout performs better than baseline MAML - maybe it's an effect of a larger number of parameters in the model?

Edit:
Thank you for your response.

I will leave my score as is.
 I would strongly encourage the authors to incorporate the baseline "(1)" as proposed by R3 in a future version of the paper as I agree with them that this is a relevant baseline.

**Experience Assessment:**

I have read many papers in this area.

**Review Assessment: Checking Correctness Of Derivations And Theory:**

I assessed the sensibility of the derivations and theory.

**Review Assessment: Checking Correctness Of Experiments:**

I assessed the sensibility of the experiments.

**Review Assessment: Thoroughness In Paper Reading:**

I read the paper at least twice and used my best judgement in assessing the paper.

---

> ### Author Response · Authors · 2019-11-11
> **Response to Reviewer #2**
>
> We sincerely appreciate your constructive comments. We respond to your main concerns below:
>
> 1. The Deterministic Meta-Dropout performs better than baseline MAML - maybe it's an effect of a larger number of parameters in the model?
>
> - To demonstrate that strong generalization performance of Meta-Dropout is not the effect of using larger number of model parameters, we doubled the number of channels for the base model and report its performances (MAML(x2)).
>
> Models		   #param.	Omni-1shot	Omni-5shot	mimg-1shot	mimg-5shot
> MAML		   x1	        	95.23+-0.17	98.38+-0.07	49.58+-0.65	64.55+-0.52
> MAML(x2)	   x4	        	94.96+-0.16	98.36+-0.08	48.19+-0.64	65.84+-0.52
> Meta-SGD         x2	        	96.16+-0.14	98.54+-0.07	48.30+-0.64	65.55+-0.56
> Meta-dropout  x2	        	96.63+-0.13	98.73+-0.06	51.93+-0.67	67.42+-0.52
>
> The number of parameters of MAML(chx2) is four times of that of MAML, while Meta-dropout is only doubled. Nonetheless, MAML(chx2) does not improve on MAML, demonstrating that the effectiveness of meta-dropout does not simply come from using larger number of parameters. Meta-SGD also doubles the number of parameters in the base MAML model, but is significantly outperformed by Meta-dropout.
>
> We want to emphasize that Deterministic meta-dropout is also one of our models, and that its good performance does not hurt our claim on the effectiveness of the multiplicative noise. This is because meta-dropout consists of two parts: meta-learned deterministic multiplicative perturbation and random noise. Thus the deterministic meta-dropout still “learns to perturb”, although not random, and is actually a core component of meta-dropout (See Table 3 in the revision). Please also see our response to the Reviewer #3, comment #4.

---

### Official Review · AnonReviewer1 · 2019-10-27
**Official Blind Review #1**

**Rating:** 6

**Review:**

This paper proposes meta dropout, which leverages adaptive dropout training for regularizing gradient based meta learning models, e.g., MAML and MetaSGD. Experiments on few shot learning show that meta dropout achieves better performance.

Overally, I think this paper is well motivated and experiments on few shot learning are impressive. I have only two major concerns.

1. Sec 3.2. According to my understanding, Meta dropout introduces a learnable prior for latent $z$, but the training objective does not require posterior inference and thus no variational inference is needed. I think it is ok to say that meta dropout tries to optimize a lower bound of log p(Y|X;\theta,\phi^*), but meta dropout does not regularize the variational framework because there is no variational inference framework.

2. Experiments on adversarial robustness can be further improved. (1) the settings and the analysis of adversarial robustness experiment can be discussed in details. For example, how to build ''adversarial learning baseline'' in meta learning settings and why the result implies the perturbation directions for generalization and robustness relates to each other; (2) how other regularization methods (e.g., Mixup, VIB and Information dropout) perform on adversarial robustness? Does Meta dropout performs better than them? (3) FGSM is a quite weak adversarial attack method, which makes evaluating adversarial robustness on FGSM may be misleading. I suggest trying some other STOA attack methods (e.g., iterative methods).

Some typos:
Page 3, Regularization methods, 3rd line, ````wwwdiscuss
Page 7, 2nd line from the bottom, FSGM->FGSM


**Experience Assessment:**

I have read many papers in this area.

**Review Assessment: Checking Correctness Of Derivations And Theory:**

I assessed the sensibility of the derivations and theory.

**Review Assessment: Checking Correctness Of Experiments:**

I assessed the sensibility of the experiments.

**Review Assessment: Thoroughness In Paper Reading:**

I read the paper at least twice and used my best judgement in assessing the paper.

---

> ### Author Response · Authors · 2019-11-11
> **Response to Reviewer #1**
>
> We really appreciate your constructive comments. We respond to each comment as follows.
>
> 1. Meta dropout does not regularize the variational framework because there is no variational inference framework.
>
> - Thank you for your comment. We agree with you that the current lower bound is not a variational form due to the assumption of q=p. In Section 3.2, we toned down the original expression “Learning to regularize variational inference“ into “Connection to variational inference”, and corrected the corresponding sentences. Still, there exists a clear connection between standard variational inference and our learning framework. Thus we believe that discussion in Section 3.2 will be helpful to readers who want to understand the meaning of learning objective Eq.(2) in depth.
>
> 2. Improving adversarial robustness experiment.
>
> - Thank you for the helpful suggestion. During the rebuttal period, we conducted additional experiments on adversarial robustness as you suggested:
>
> a) We replaced the previous FGSM attack with stronger PGD attack (200 iter.), with $L_1$, $L_2$, and $L_\infty$ norm constraints.
>
> b) We included more baselines (e.g. Mixup, VIB, and information dropout), and show that our meta-dropout largely and consistently outperforms all of them.
>
> c) We added more detailed descriptions of the adversarial meta-learning baseline and in-depth analysis on the results.
>
> d) We further show that the learned perturbation from our Meta-dropout also generalize across different types of adversarial attacks with $L_1$, $L_2$, and $L_\infty$ attacks. The generalization to different types of attacks is an important problem in adversarial learning, and most existing models fail to achieve this goal.
>
> Please see the corresponding section in the revision. We believe that the adversarial robustness part of our paper has become much stronger than before, thanks to your suggestion.

---

### Author Response · Authors · 2019-11-11
**Summary of updates in the revision**

We thank all reviewers for their constructive comments. Here we briefly mention what have been updated in the revision. For more detailed explanations, please refer to the response to each reviewer.

1. New experimental results on adversarial robustness: we replaced the FGSM attack with PGD attacks with $L_1$, $L_2$, $L_\infty$ norm, and included more baselines (e.g. Mixup, VIB, and Information Dropout). The results show that our meta-dropout yields deep neural networks that are significantly more robust to adversarial attacks than baselines, regardless of the types of attacks used ($L_1$, $L_2$, $L_\infty$) on few-shot classification tasks. We believe that these results will significantly strengthen our work, as we have shown that our meta-dropout not only achieves models that generalize better, but are more robust. While these findings are from few-shot classification tasks, we believe that they are meaningful, since existing models for adversarial learning achieved robustness at the expense of generalization accuracy and most of them do not generalize across different types of attacks. Please see the paragraph “Adversarial robustness” and Figure 5 in Page 7, as well as our response to the R1’s comment below for more detailed explanations.

2. In Section 3.2, based on R1’s comment, we toned down on our claims. We changed “Learning to regularize variational inference“ into “Connection to variational inference” and corrected corresponding sentences that look overstated.

3. In Table 3, for each baseline, we added in the indication on whether it has the following properties: Random sampling, Learned multiplication, and Input-dependency. This allows us to better see the effect of each component to the generalization performance. We also denote that the input-dependent meta-learning of multiplicative noise with stochasticity as the core components of our meta-dropout. We provide more in-depth analysis on “Ablation study” paragraph in Page 8.

---

### Decision · Program_Chairs · 2019-12-19

**Decision:**

Accept (Poster)

**Comment:**

This paper proposes a type of adaptive dropout to regularize gradient based meta-learning models. The reviewers found the idea interesting and it is supported by improvements on standard benchmarks. The authors addressed several concerns of the reviewers during the rebutal phase. In particular, revisions added results against other regularization mthods. We recommend that further attention is given to ablations, in particular the baseline proposed by Reviewer 1.